# Evaluation of an Antifungal Stewardship Initiative Targeting Micafungin at an Academic Medical Center

**DOI:** 10.3390/antibiotics12020193

**Published:** 2023-01-17

**Authors:** J. Myles Keck, David A. Cretella, Kayla R. Stover, Jamie L. Wagner, Katie E. Barber, Tulip A. Jhaveri, Prakhar Vijayvargiya, Zerelda Esquer Garrigos, Mary Joyce B. Wingler

**Affiliations:** 1Department of Pharmacy, University of Arkansas for Medical Sciences, Little Rock, AR 72205, USA; 2Division of Infectious Diseases, University of Mississippi Medical Center, Jackson, MS 39216, USA; 3Department of Pharmacy Practice, University of Mississippi School of Pharmacy, Jackson, MS 39216, USA

**Keywords:** echinocandin, invasive candidiasis, stewardship, pharmacy, infectious diseases

## Abstract

Delays in the treatment of proven invasive fungal disease have been shown to be harmful. However, empiric treatment for all patients at risk of infection has not demonstrated benefit. This study evaluates the effects of a micafungin stewardship initiative on the duration of therapy and clinical outcomes at the University of Mississippi Medical Center in Jackson, Mississippi. This single-center quasi-experiment evaluated patients who received micafungin. Adult inpatients who received at least one treatment dose of micafungin in the pre-intervention (1 October 2020 to 30 September 2021) or post-intervention (1 October 2021 to 30 April 2022) groups were included. Patients were placed on micafungin for prophylaxis and those who required definitive micafungin therapy were excluded. An algorithm was used to provide real-time recommendations in order to assess change in the treatment days of micafungin therapy. A total of 282 patients were included (141 pre-group versus 141 post-group). Over 80% of the patients included in the study were in an intensive care unit, and other baseline characteristics were similar. The median number of treatment days with micafungin was 4 [IQR 3-6] in the pre-group and 3 [IQR 2-6] in the post-group (*p* = 0.005). Other endpoints, such as time to discontinuation or de-escalation, hospital mortality, and hospital length of stay, were not significantly different between the groups. An antifungal stewardship initiative can be an effective way to decrease unnecessary empiric antifungal therapy for patients who are at risk of invasive fugal disease.

## 1. Introduction

Over the last two decades, invasive fungal infections involving *Candida* spp. have risen, leading to an increase in the empiric use of antifungal therapies [1]. However, the overuse of antifungals has led to the promotion of resistance without clinical benefit [2,3,4,5,6,7,8,9]. To prevent unnecessary antifungal therapy and subsequent resistance to these agents, providers utilize a diagnostic approach that commonly involves assessing patient-specific risk factors, such as central lines, clinical presentations, and diagnostic tests, most commonly blood cultures and β-1,3-D-glucan (BDG) [10,11,12,13,14,15,16,17]. Unfortunately, both blood cultures and BDG have limitations, making their utility in certain clinical scenarios difficult [10,11,12]. For instance, BDG has utility in deep-seated infections, but variable specificity and sensitivity in critically ill patients [12,13,15,16]. In contrast, blood cultures have become the gold standard for candidemia, yet have limited utility in deep-seated infections [15,16,17]. Together, these tests highlight both the drawbacks of fungal diagnostics and the diagnostic challenges for patients who are at risk of invasive candidiasis (IC).

Studies have shown that empiric therapy for patients at high risk of IC have failed to improve patient outcomes [18,19,20], yet delayed antifungal therapy is harmful in patients with confirmed IC [21]. This creates a challenge where urgent empiric therapy may have been life-saving in those eventually diagnosed with candidemia, but the empiric treatment of all patients presumably at high risk of IC is not beneficial. Current guidelines recommend the empiric use of antifungals when IC is suspected, or if the patient does not clinically improve on empiric antibiotic therapy [22,23]. However, there is increased concern for the growing resistance seen in non-*albicans* species against micafungin, thus promoting the need for better antifungal stewardship practices moving forward [24,25,26,27]. Unfortunately, antifungal stewardship presents many challenges that are not often seen in traditional antimicrobial stewardship initiatives [28]. These challenges center around the fact that empiric antifungal therapy for IC is commonly employed in critically ill patient populations [29,30]. Historically, stewardship in the critically ill has not been a “low-hanging” intervention, especially in the initial stages of stewardship programs [31,32]. As rates of antifungal resistance continue to rise, clinical practices involving antifungal stewardship should become more commonplace in institutions [33]. In 2020, the Infectious Diseases Society of America (IDSA) released a supplemental article which outlined how the core elements of stewardship could be utilized in the setting of antifungal stewardship [28]. Specifically, the authors suggest a multidisciplinary approach to antifungal stewardship that consists of an infectious diseases (ID) physician and a pharmacist collaborating in the management of patients at risk of IC [28]. Multiple studies have generated evidence demonstrating the benefits of antifungal stewardship [34,35,36,37].

The purpose of this study was to evaluate the impact of an antifungal stewardship initiative on the empiric use of micafungin within our institution.

## 2. Results

A total of 548 patients were screened and 282 patients were included (141 pre-intervention, 141 post-intervention). The most common reasons for exclusion included hematologic malignancy (including acute myeloid leukemia (AML) and acute lymphocytic leukemia (ALL)) and QTc prolongation (>500 ms) or elevated aspartate transferase (AST)/alanine aminotransferase (ALT) five times the upper limit of normal precluding the use of fluconazole. Serum creatinine was slightly higher in the post-intervention group, otherwise baseline demographics did not differ between the two groups. Over 80% of patients were admitted to the intensive care unit (ICU) at the time of micafungin initiation, and over 50% were on vasopressors and/or mechanical ventilation (Table 1). The remaining 20% of patients were admitted to general ward services.

The median days of therapy with micafungin was 4 days [interquartile range (IQR) 3,6] in the pre-intervention group compared to 3 days [IQR 2,6] in the post-intervention group (*p* = 0.005). The secondary outcomes, including length of stay (LOS), in-patient mortality, and micafungin discontinuation or de-escalation, did not differ between the two groups (Table 2). The *Candida* species isolated from blood (*C. albicans* (13), *C. parapsilosis* (4), and *C. tropicalis* (3)) and wound (*C. albicans* (17), *C. glabrata* (4), *C. parapsilosis* (2), and other (1)) cultures were all susceptible to fluconazole. The median time to discontinuation from the 72-h time out was 1 day in the pre-group [IQR 0, 2] vs. 1 day in the post-group [IQR 0, 3] (*p* = 0.144). The median duration of time to de-escalation from the 72-h time out was 1 day in the pre-group [IQR 0, 3.5] vs. 0 days in the post-group [IQR 0, 3] (*p* = 0.603).

Early adherence to diagnostic recommendations was evaluated. Within 24 h of micafungin initiation, similar rates of patients in the pre- and post-groups had blood cultures (72% vs. 74%, *p* = 0.893) and BDG (52% vs. 53%, *p* = 0.812) obtained. Ten patients (7%) in both groups had positive blood cultures for *Candida* (*p* = 1.000). Positive wound cultures for *Candida* were present in 13 (9%) compared with 11 (8%) (*p* = 0.832) patients in the pre- and post-intervention groups, respectively. Most patients had BDG performed, and the BDG results took significantly longer to return in the post-intervention group. The BDG results were positive in 21% and 35% of patients in the pre- and post-group, respectively (*p* = 0.068). Repeat BDGs were performed in 33% and 15% of patients with an initial positive BDG in the pre- and post-group, respectively (*p* = 0.248). Of these, 80% and 50% were positive in the pre- and post-group, respectively. The rates of candidemia were relatively consistent, with the exception of an increase during August–September 2021. However, micafungin use fluctuated over time (Figure 1).

## 3. Discussion

This evaluation of an antifungal stewardship initiative demonstrated a reduction in the median days of micafungin therapy after the implementation of a multidisciplinary stewardship intervention, which included continual audit and feedback along with multidisciplinary education sessions with ICU medical teams. The time to de-escalation of therapy was numerically shorter in the post-group (1 day vs. 0 days, *p* = 0.603), demonstrating another positive outcome of this initiative.

Other studies have evaluated the impact of antifungal stewardship interventions. In 2012, Lopez-Medrano and colleagues initiated a 12-month stewardship program with daily prospective audits and feedback regarding the use of oral voriconazole, intravenous voriconazole, caspofungin, and liposomal amphotericin B. At the end of the 12 month period, the defined daily doses (DDDs) of intravenous voriconazole and amphotericin B were reduced by 31% and 20.2%, respectively, which amounted to cost savings of over $30,000 [35]. In 2013, Mondain and colleagues evaluated the impact of a 6-year stewardship program on antifungal usage. From 2005 to 2010, 344 of the 636 prescriptions had a stewardship intervention performed (diagnostic workup and antifungal therapy selection), which led to the containment of antifungal therapy expenses (*p* = 0.015) and higher rates of appropriate first-line therapy for confirmed candidemia (*p* = 0.0025) over the 6 year period [37]. Lastly, in 2015, Micallef and colleagues evaluated a stewardship program which targeted echinocandins, voriconazole, and/or liposomal amphotericin B. During a 12-month period, 40.8% of liposomal amphotericin B, 18% voriconazole, 72.2% caspofungin, and 64.7% of micafungin prescriptions were stopped or changed to a more focused antifungal agent, resulting in a total saving of over £170,000 [36]. Though our study did not directly evaluate expense-related outcomes, it is expected that cost savings occurred due to decreased micafungin usage.

In this study, empirical antifungal therapy did not impact clinical outcomes (inpatient mortality and length of hospital stay). Although this population included a wide range of patients, the majority of included patients were in an intensive care unit (ICU). Two studies assessed similar populations [38,39]. A randomized trial by Schuster and colleagues evaluated the role of empiric fluconazole versus a placebo in patients admitted to the ICU. At 4 weeks post-therapy, 36% vs. 38% (*p* = 0.78) of the fluconazole and placebo group patients had successful outcomes, respectively [38]. The EMPIRICUS study by Tomsit and colleagues in 2016 examined whether empiric micafungin therapy could improve fungal infection-free survival at day 28 in ICU patients. Fungal infection-free survival at day 28 was seen in 68% and 60.2% (*p* = 0.18) of patients who received micafungin and a placebo, respectively [39]. Collectively, these results point to similar clinical outcomes regardless of the addition of empiric coverage for IC, even in select critically ill ICU patients, and the need for antifungal stewardship to circumvent potential rises in antifungal therapy resistance due to the overuse of these agents [40,41].

Evaluating the laboratory tests and microbiologic cultures obtained from the patients included in this study was helpful to identify areas for improvement in the future. For example, despite encouraging limited usage of BDG for patients without a deep-seated infection, 70% of the patient population had a BDG performed during the micafungin course evaluated. Since BDG return times are variable at our institution, the discontinuation of therapy was often delayed while awaiting these results. Furthermore, of the initial BDG performed, only 21% and 35% were positive for the pre- and post-group, respectively, highlighting the poor overall sensitivity and specificity of BDG in this patient population. On the other hand, our recommendations suggested that patients with a positive BDG should have a repeat performed. In these patients, 80% and 50% were positive in the pre- and post-group, respectively. These results mirror those seen in previous studies where repeat BDGs increase the sensitivity in critically ill patient populations [17,42]. In our study, a higher numerical percentage of initial BDGs were positive in the post-group. Though not fully illustrated in the results, we believe this result was driven by a higher number of intra-abdominal surgery patients, higher rates of candidemia due to COVID-19 variant outbreaks, and the proper utilization of BDGs for patients with deep-seated infections. However, other causes cannot be excluded. A limitation of our study is that the prospective audit with feedback was generally targeted at micafungin rather than the diagnostic stewardship of BDG since most of the work-up was ordered prior to the patient being reviewed. This should be considered in other programs that wish to implement antifungal stewardship.

Inversely to BDG, blood cultures were encouraged for all the patients in our study prior to micafungin initiation. However, only 75% of patients had a blood culture performed within 24-h of micafungin initiation. Blood cultures have good sensitivity to detect candidemia and are not prone to false positives, unlike BDG [43,44,45,46,47,48]. In addition, all the positive blood cultures are reviewed by our antimicrobial stewardship and critical care pharmacists, and our institution uses rapid diagnostics to assist with the identification of yeast in blood cultures. Antifungal therapy is quickly initiated or modified based on those results, which can positively impact the time to appropriate therapy. A previous study demonstrated improved antifungal use with an intervention that targeted early responses to candidemia [49]. Improving blood culture collection in patients with suspected IC will be a high priority in future interventions at our institution.

As stated previously, critically ill patients were the predominant population in this study and present many challenges commonly unseen in traditional stewardship activities [50,51]. In addition, during the study period, ICU providers at our institution became hesitant to engage in stewardship activities due to a host of factors, including COVID-19 variant waves conferring more severe outcomes, delayed BDG return times, and increased mortality rates in critically ill patients throughout the COVID-19 pandemic [52]. This trend was educational, as environmental stressors, specifically COVID-19, can significantly impact clinical decision-making. Antimicrobial stewardship programs should determine the best way to approach providers who have changed antibiotic prescribing practices during the pandemic. One suggestion is to implement handshake stewardship, which is what our institution has recently returned to following a pause throughout most of 2020 and 2021.

Our study has several limitations. First, this is a single-center, retrospective, small population study, which limits the external validity. Additionally, recall and selection bias are common in retrospective studies, although an extensive data collection process was undertaken to minimize these biases. Moreover, only the first occurrence (i.e., first time receiving micafungin) was documented, and subsequent micafungin initiations for patients with long-term stays were not documented. As a result, true micafungin usage could be much greater within our institution as many patients with COVID-19 were placed on micafungin subsequent times during their admission and these data were not tracked. As briefly mentioned, during both the pre- and post-group time periods, multiple COVID-19 variants surfaced. This generated numerous stewardship challenges as the acquirement of COVID-19 was associated with an increased risk of IC in our hospital (Figure 1). Reactively, providers at our hospital had a small threshold to escalate anti-infective therapy in critically ill patients with continual decline (Figure 1). This led to the tripling of rates of echinocandin usage during COVID-19 surges within our hospital. Furthermore, the providers within our institution became heavily dependent on the BDG results before they considered discontinuing micafungin therapy during the COVID-19 pandemic. Unfortunately, BDG is a send-out test for our institution with variable return times, further hindering the ability to do effective prospective audits and feedback. In addition, patients prescribed micafungin were evaluated three to four times per week with no weekend coverage, which could have led to missed prospective audit and feedback opportunities. Finally, we cannot fully exclude other confounding clinical factors driving this result, such as higher rates of micafungin initiation prior to patient expiration and/or further therapy escalation to anti-mold agents.

## 4. Materials and Methods

This single-center, pre-/post-intervention quasi-experimental study evaluated patients admitted to the University of Mississippi Medical Center in Jackson, Mississippi who received micafungin, the echinocandin on the institution’s formulary. Patients were identified using TheraDoc^®^ Clinical Surveillance Software (Premier^®^, Charlotte, NC, USA). The pre-intervention group included patients who were admitted from 10/01/2020 to 09/30/2021. The post-group included patients who were admitted from 10/01/2021 to 04/31/2022. All data was collected using REDCap [53]. This study was approved by the University of Mississippi Medical Center Institutional Review Board (protocol number #2021-1052).

### 4.1. Study Intervention

A pharmacy-driven micafungin antimicrobial stewardship initiative was created alongside the Division of ID (Figure 2). Blood cultures were recommended for all patients, and a BDG was recommended for patients with recent intra-abdominal surgery. A 72-h time out was recommended, at which time the providers would determine whether micafungin should be discontinued or de-escalated, or if additional work-up was required based on the blood culture and/or BDG results. BDG is a send-out lab at our institution with variable turnaround times (2–4 days).

Medical teams, specifically the critical care units with historically high micafungin usage, were educated on the micafungin intervention in September 2021. The antimicrobial stewardship team, which consisted of two antimicrobial stewardship and ID trained pharmacists and one ID pharmacy resident, reviewed patients on micafungin three times a week as part of normal stewardship activities from October 2021 through April 2022. Recommendations, which consisted of antifungal de-escalation, antifungal discontinuation, and ID consult, were provided via prospective audits and feedback.

### 4.2. Inclusion and Exclusion Criteria

Patients who were 18 years or older who had received at least one treatment dose of micafungin during their hospital admission were included. Patients who required definitive echinocandin therapy for culture-confirmed infection (*Candida* spp. resistant to fluconazole, combination therapy for mold infection, QTc > 500, AST/ALT > 5x the upper limit of normal (ULN)) or those who received micafungin for prophylaxis were excluded. Prophylaxis was used as per institutional protocol in individuals with an absolute neutrophil count (ANC) of less than 500 cells/mm^3^ with the diagnosis of a hematologic malignancy (defined as any cancer involving the blood, bone marrow, and lymph nodes, including, but not limited to, acute lymphocytic leukemia (ALL), chronic lymphocytic leukemia (CLL), acute myeloid leukemia (AML), chronic myeloid leukemia (CML), multiple myeloma, Hodgkin’s lymphoma, and non-Hodgkin’s lymphoma (NHL)), solid-organ transplant (lung, liver, heart, or kidneys), or stem cell transplant (autologous and allogeneic).

### 4.3. Outcomes

The primary outcome was the median number of days of micafungin therapy. The secondary outcomes evaluated included hospital LOS, in-patient mortality, time to de-escalation after the 72-h time out period, time to discontinuation after the 72-h time out period, the percentage of patients with a BDG obtained within 24-h of micafungin initiation, the percentage of patients with a BDG ordered during the micafungin course evaluated, and the rates of confirmed candidemia.

### 4.4. Statistical Methods

All statistical analyses were performed using SPSS Software, version 28 (Armonk, NY, USA: IBM Corp). All the categorical variables were analyzed using a Fisher’s exact test. All the continuous variables were analyzed using a Mann–Whitney U test. Power for the study was determined using an online power calculator (ClinCalc). It was determined that 141 patients were needed in each group in order to meet a power of 80%. In the pre-group, the list of patients was pulled retrospectively and was randomized. Data were collected in a convenience sample until the desired sample size was met. The *p*-value was set as <0.05, which inferred statistical significance for the primary outcome.

## 5. Conclusions

In this quasi-experimental study, a micafungin stewardship initiative was implemented and led to a decreased duration of micafungin treatment. These results demonstrate the potential implications of an antifungal stewardship initiative to decrease unnecessary empiric antifungal therapy in patients at risk of IC. These interventions could be applied to other antifungals or echinocandins, depending on the opportunities for improvement identified at each individual institution. Antifungal stewardship programs should also consider incorporating diagnostic stewardship interventions, specifically with regard to BDG.

## Figures and Tables

**Figure 1 antibiotics-12-00193-f001:**
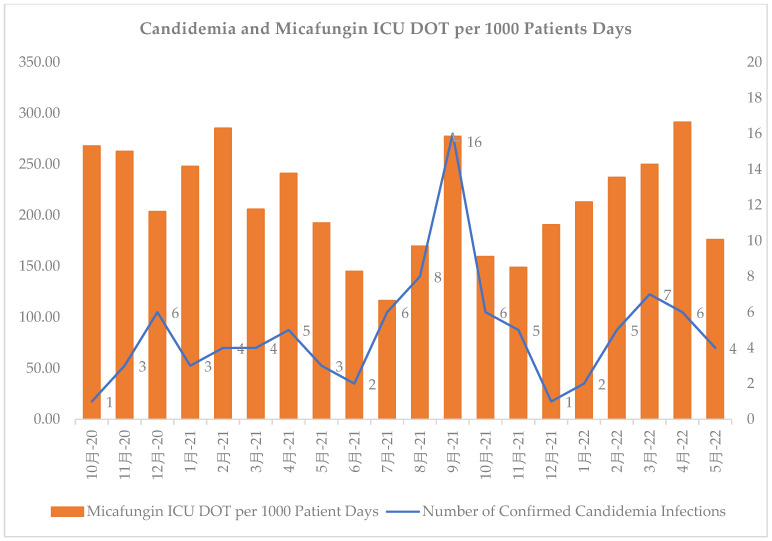
Number of confirmed cases of candidemia vs. days of micafungin therapy from October 2020 to May 2022 at an academic medical center. Chinese in figure is “month”.

**Figure 2 antibiotics-12-00193-f002:**
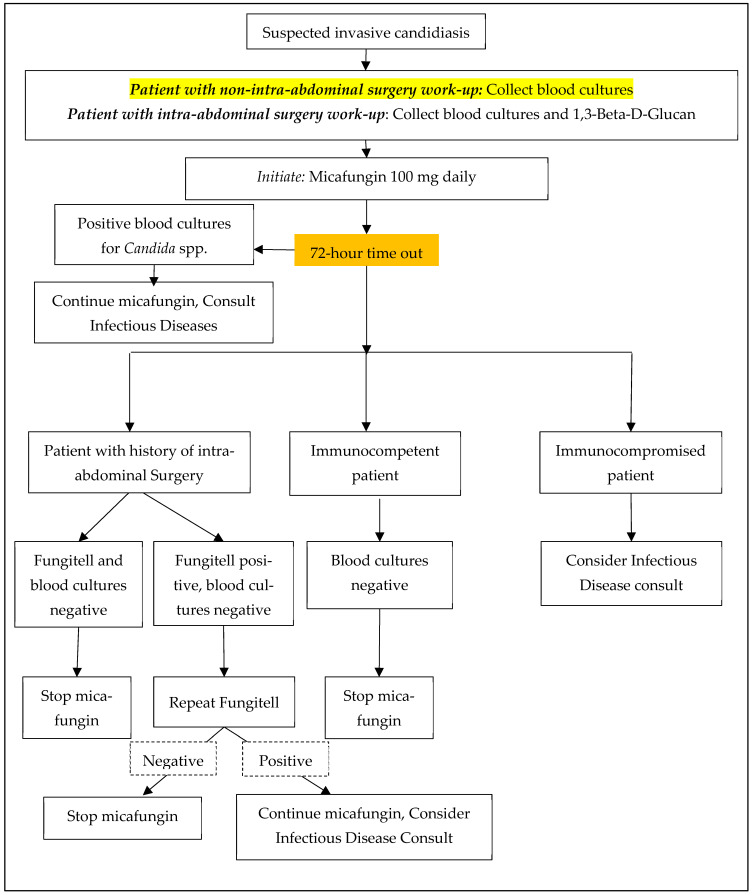
Micafungin Antifungal Stewardship Algorithm.

**Table 1 antibiotics-12-00193-t001:** Baseline characteristics of included participants.

Variable Median (IQR) or n (%)	Pre-Intervention Group (n = 141)	Post-Intervention Group (n = 141)	*p*-Value
Age (years)	54 (41, 66)	58 (42, 66)	0.337
Weight	84 (68, 104)	81 (61, 102)	0.141
Serum creatinine ^a^	1.3 (0.54, 2.40)	1.7 (0.61, 2.7)	0.038
Charlson Comorbidity Index	3 (1, 5)	3 (1, 5)	0.293
Fungal culture ^b^ positive for *Candida* spp. (previous 12 months)	10 (7)	5 (4)	0.288
History of Cancer	35 (25)	24 (17)	0.143
ICU admission ^a^	113 (80)	117 (83)	0.645
Abdominal surgery	33 (23)	26 (18)	0.380
Parenteral nutrition ^a^	10 (7)	16 (11)	0.303
Mechanical ventilation ^a^	83 (59)	83 (59)	1
Vasopressors ^a^	75 (53)	80 (57)	0.632

^a^ At the time of micafungin initiation; ^b^ any blood, sputum, or urine cultures; IQR: interquartile range; ICU: intensive care unit.

**Table 2 antibiotics-12-00193-t002:** Primary and Secondary Outcomes.

Outcome Median (IQR) or n (%)	Pre-Intervention Group (n = 141)	Post-Intervention Group (n = 141)	*p*-Value
Micafungin days of therapy	4 (3, 6)	3 (2, 6)	0.005
Hospital Length of Stay, days	28 (12.5, 45)	22 (12, 38)	0.137
In-patient mortality	64 (45)	68 (48)	0.634
Confirmed *Candida* infection ^a^	20 (14)	21 (15)	1.000
Micafungin adjustments ^b^ at 72 h Discontinuation De-escalation	80 (66%) 11 (55%)	77 (64%) 14 (67%)	0.788 0.530
Beta-D-Glucan performed	100 (71)	94 (67)	0.521
>1 Beta-D-Glucan	36 (26)	18 (13)	0.001
Time to Beta-D-Glucan results, days	2 (2, 3)	3 (2, 4)	<0.001

^a^ Three patients in the pre-group had a positive blood and wound culture; all the other patients had either a positive blood or wound culture for *Candida* spp. No *Candida* species resistant to fluconazole were included. ^b^ Discontinuation was measured in patients without an identified Candida infection. De-escalation was evaluated in patients with a confirmed fluconazole-susceptible Candida infection. IQR: interquartile range.

## Data Availability

Not applicable.

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
