# Peer review of "Evaluation of an Antifungal Stewardship Initiative Targeting Micafungin at an Academic Medical Center"

_antibiotics, 2023, doi:10.3390/antibiotics12020193_

Round 1

Reviewer 1 Report

Abstract

1.      Please rephrase the first sentence

2.      Line # 16 write name and location of medical center

3.      Line # 22, sentence not start from numerical

4.      Line # 24, 1QR (3-6) and (2-6)

Introduction

1.      Line # 53, reference # 24, your study is related to only micafungin, why are you mentioning all echinocandins? it will be better if you mention micafungin only, you may add this literature http://dx.doi.org/10.1080/21505594.2022.2123325.

Results

1.      Line # 76, over 80% were ICU admitted patients, what about the other 20%, are they outpatients or admitted in other departments, please specify.

2.      Please specify the Candida species name and % of resistant to each species type. Normally the albican are less resistant than non-albican.

Discussion

1.      Please discuss the nosocomial transmission of candida in hospital seating, which is an important factor for dissemination of MDR candida

2.      Please mention this literature in your discussion section

 https://doi.org/10.3390/antibiotics10101217

http://dx.doi.org/10.3389/fmicb.2022.981181

3.      Please discuss the implications of your study.

Method

1.      Please specify the study center name and location

Author Response

Abstract

  1. Please rephrase the first sentence

We have rephrased to:

Delays in treatment of proven invasive fungal disease have shown to be harmful. However, empiric treatment for all patients at risk of infection has not demonstrated benefit.

  1. Line # 16 write name and location of medical center

Added.

  1. Line # 22, sentence not start from numerical

Revised to “A total of 282 …”

  1. Line # 24, 1QR (3-6) and (2-6)

Revised as suggested.

Introduction

  1. Line # 53, reference # 24, your study is related to only micafungin, why are you mentioning all echinocandins? it will be better if you mention micafungin only, you may add this literature http://dx.doi.org/10.1080/21505594.2022.2123325.

We have changed this to micafungin only, and added the reference as suggested.

Results

  1. Line # 76, over 80% were ICU admitted patients, what about the other 20%, are they outpatients or admitted in other departments, please specify.

We have added the following sentence:

The remaining 20% of patients were admitted to general ward services.  

  1. Please specify the Candida species name and % of resistant to each species type. Normally the albican are less resistant than non-albican.

We have added the species breakdown above Table 2, and added a footnote to Table 2 to report that no patients with Candida species resistant to fluconazole were included.

Discussion

  1. Please discuss the nosocomial transmission of candida in hospital seating, which is an important factor for dissemination of MDR candida

Thank you for this suggestion. While we agree that this is true, this study focused only on susceptible isolates. We feel that a deep dive into nosocomial transmission and MDR Candida deviates from the purpose of the article and may be confusing to the readers, so we have elected not to include this.  

  1. Please mention this literature in your discussion section

 https://doi.org/10.3390/antibiotics10101217

http://dx.doi.org/10.3389/fmicb.2022.981181

We have added the first citation as suggested. We feel that the second is less directly related and have elected to leave that one out.   

  1. Please discuss the implications of your study.

We have added several statements to speak to implications (discussion lines 153-155, 172-173, 191-192, and 211-215) and revised the conclusion to the following:

In this quasi-experimental study, a micafungin stewardship initiative was implemented and led to a decreased duration of micafungin treatment. These results demonstrate the potential implications of an antifungal stewardship initiative at decreasing unnecessary empiric antifungal therapy in patients at risk for IC. These interventions could be applied to other antifungals or echinocandins, depending on the opportunities for improvement identified of each individual institution. Antifungal stewardship programs should also consider incorporating diagnostic stewardship interventions, specifically with regard to BDG.

Method

  1. Please specify the study center name and location

We have revised as suggested. 

Reviewer 2 Report

This is an interesting study entitled Evaluation of a micafungin antifungal stewardship initiative at an academic medical center”. Studies involving antifungal use and role of pharmacist are important. Results from these studies are essential in implementing antimicrobial stewardship activities. The passion of the authors to improve antifungal use is acknowledged. This paper has a potential to be accepted, but few points must be explained before going forward.

·         I’d suggest to change title because it is bit confusing

·         The paper is well written with good collection of data; however, authors are requested to collect and add more information from the papers/literature that are published after 2020. The article lacks in this area.

·         I’d suggest to use abbreviations like ns (non sig), * (sig) and ** (highly sig) instead of writing p values.  

·         Globally I suggest you write your article towards the international readership of your targeted journal.

·         The conclusions must be drawn appropriately based on the data presented. 

Author Response

  • I’d suggest to change title because it is bit confusing

We have revised to “Evaluation of an antifungal stewardship initiative targeting micafungin at an academic medical center”.

  • The paper is well written with good collection of data; however, authors are requested to collect and add more information from the papers/literature that are published after 2020. The article lacks in this area.

We have added several new citations to address this point.

  • I’d suggest to use abbreviations like ns (non sig), * (sig) and ** (highly sig) instead of writing p values.  

Thank you for this suggestion. We define significance in the methods, so have elected to leave p-values included as currently written.

  • Globally I suggest you write your article towards the international readership of your targeted journal.

Thank you for the suggestion. We have made edits throughout to improve readability.

  • The conclusions must be drawn appropriately based on the data presented. 

We have revised the conclusion to read as follows:

In this quasi-experimental study, a micafungin stewardship initiative was implemented and led to a decreased duration of micafungin treatment. These results demonstrate the potential implications of an antifungal stewardship initiative at decreasing unnecessary empiric antifungal therapy in patients at risk for IC. These interventions could be applied to other antifungals or echinocandins, depending on the opportunities for improvement identified of each individual institution. Antifungal stewardship programs should also consider incorporating diagnostic stewardship interventions, specifically with regard to BDG.